# Implications of Spatio-Temporal Land Use/Cover Changes for Ecosystem Services Supply in the Coastal Landscapes of Southwestern Ghana, West Africa

**Stephen Kankam [1,2,*], Adams Osman [3], Justice Nana Inkoom [1] and Christine Fürst [1,4]**

1 Department of Sustainable Landscape Development, Institute for Geosciences and Geography, Martin Luther University Halle-Wittenberg, Von-Seckendorff-Platz 4, 06120 Halle, Germany
2 Hen Mpoano (Our Coast), 38. J Cross Cole Street, Windy Ridge Extension, Takoradi P.O. Box AX 296, Ghana
3 Department of Geography Education, University of Education, Winneba P.O. Box 25, Ghana
4 German Centre for Integrative Biodiversity Research (iDiv) Halle-Jena-Leipzig, Puschstraße 4, 04103 Leipzig, Germany
* Correspondence: stephen.kankam@student.uni-halle.de

**Abstract:** Land use/land cover change (LULCC) is an important driver of ecosystem changes in coastal areas. Despite being pervasive in coastal Ghana, LULCC has not been investigated to understand its effects on the potential for coastal landscapes to supply ecosystem services (ES). In this study, the impacts of LULCC on the potential supply of ES by coastal landscapes in Southwestern Ghana was assessed for the years 2008 and 2018 by using remote sensing and benefit transfer approaches. Based on available data, relevant provisioning and regulating ES were selected for the assessment while indicators to aid the quantification of the ES were obtained from literature. Supervised classification methods and maximum likelihood algorithms were used to prepare land use/land cover (LULC) maps and the derived LULC categories were assigned according to the descriptions of the Land Cover Classification System (LCCS). Potential supply of provisioning (food, fuelwood) and regulating (carbon storage) services was quantified and the spatial and temporal distributions of these ES illustrated using maps. The results show variations in food and fuelwood supply and carbon storage potentials over the study period and across different locations on the landscape. Potentials for fuelwood supply and carbon storage in mangrove forests indicated declining trends between 2008 and 2018. On the other hand, food-crop supply and carbon storage potential in rubber plantations depicted increasing patterns over the same period. Population, slope and elevation exhibited strong effects on LULC conversions to food crop and rubber plantations whereas these factors were less important determinants of mangrove forest conversions. The findings of the study have implications for identifying and addressing tradeoffs between land uses for agriculture, industrial development and conservation of critical coastal ES within the context of rapid land system transformations in the study region.

**Keywords:** ecosystem services; land use/land cover change; benefit transfer; coastal landscapes; quantification; spatio-temporal; West Africa; Ghana

## 1. Introduction

During the past half century, coastal zones have witnessed unprecedented transformation, due in part to the increasing impacts of human activities in these regions [1–3]. Urbanization patterns, natural resources exploitation, infrastructure development, industrial and commercial activities are concentrated on a narrow strip of land in the coastal zone [3]. In West Africa, rapid land use/land cover changes (LULCC) have gained prominence in the coastal zone as associated demographic, socioeconomic, technological and political drivers of change interact within the coastal socioecological system [4–6]. The accelerating pace of land use change coupled with the resulting impacts on coastal ecosystems

has heightened concerns among planners, policy-makers and scientists about the sustainability of coastlines and associated ecosystem services (ES). Research on land use/land cover dynamics and related impacts on coastal ES is therefore gaining traction in scientific discourse (e.g., [6–9]). Global-scale assessments estimate annual ES losses due to land use changes at approximately $20.2 trillion [10,11]. Increasingly, individuals and societies respond to opportunities created by globalization processes, including market conditions by altering land uses [12,13]. These changes trigger degradation and conversion of high-value ecosystems such as forests, cropland, water and grasslands to low-value land uses [10,13]. Over the last two decades, ecosystem degradation have heightened due to an exponential increase in population and doubling of economic activities with attendant increase in the demand for ecosystem goods and services [2,4]. In many developing countries, weak or absent land use regulatory institutions are critical among the conditions giving rise to rapid modifications of ecosystems and landscapes [12]. While focusing assessments on Africa, the Intergovernmental Science-Policy Platform on Ecosystem Services (IPBES) identifies significant risks of ES and biodiversity losses in the face of poorly regulated and unregulated LULCC in the continent [14].

Nevertheless, there is also universal recognition that impacts of LULCC on ES are not always negative as society–nature interactions are integral to the processes of ES coproduction [15,16]. Indeed, landscapes' potential to supply food, fiber and fodder, and perform other functions such as climate regulation are inexorably linked to LULCC [17]. Thus, availability of land use/land cover data and related ES information underpin landscape planning and land use decision-making to sustain ES [18]. Decisions to protect ecosystems are taken expeditiously and aided by the availability of low-cost information [19]. However, application of field-based measurements for biophysical data acquisition and information gathering on ES is expensive and time consuming. Additionally, while field measurements are conducted at the local scale, land use decisions to protect ES are made at relatively larger scales [19]. In West Africa, challenges associated with lack of data at the appropriate planning scale and resolution hinder the conduct of ES studies and also their integration into land use decisions [20–22]. Where there is a paucity of data on specific ES, proxies have been utilized for mapping broad-scale trends in ES supply [23,24]. Applications of such proxy-based techniques for quantifying and mapping ES involve benefits transfer [25,26]. In benefits transfer, biophysical measures or economic estimates from a previously estimated site are extrapolated to another site with similar conditions [27–29]. A key element of benefit transfer applications is homogeneity in land cover characteristics between the site from where data is transferred and the site to which the transfer data is applied [18,19]. Thus, the existence of homogenous land cover types between sites facilitates the transfer of data to aid ES quantification [30]. Yet, such proxy-based data transfers result in error propagation from the transfer site [24,31]. Relatedly, benefit transfer applications enable the use of single point estimates or average values as a basis to transfer empirical data from one site in order to estimate ES values for another site [18,20]. Provisioning ES is amenable to quantification using benefits transfer as they represent long-standing economic sectors and traditional research areas such as agriculture, fisheries and forestry, for which a large body of datasets are available [19]. Similarly, "carbon sequestration", as a regulating ES, provides opportunities for quantification using biophysical units.

Quantification of ES is a useful process for raising awareness and providing insights about critical ecosystems in land use and spatial planning systems. However, in coastal areas, land use and spatial planning systems are challenged by complex and interrelated drivers of ecosystem changes. Particularly in the coastal landscapes of Southwestern Ghana, LULCC are consequences of an evolving oil and gas industry in the marine and coastal zones, population growth, urbanization and plantation agriculture development [32,33]. Increasingly, land losses from oil palm, cropland and shrubland favor gains in rubber plantation [34]. Nonetheless, rubber plantations are fragmented over the landscape as their establishment on few acres of land are determined by individual land owners in the context of an outgrower scheme [34]. Over the past decade, the region has been the focus

of spatial planning and ecosystem-based management initiatives due to its importance for conservation and maintenance of a healthy small-scale fishery. Furthermore, the area is being explored for its potential contribution toward national climate change mitigation strategies such as REDD+ and other voluntary carbon offset programs [35]. However, with the advent of offshore oil and gas discovery in 2007, competition among land uses in Southwestern Ghana has intensified [33,36].

Recent impact assessments of LULCC in this region have focused analysis on the capacity for landscapes to supply cultural ES using participatory land use scenarios (e.g., [37]). Similar studies in the region also explored the provision of fisheries-related ES in support of decisions to establish marine protected areas (e.g., [38]). Nevertheless, the potential supply of provisioning and regulating ES by the coastal landscapes remains poorly understood. Meanwhile, such understandings are necessary to improve land use actors' awareness of potential critical losses of ES and of opportunities to sustain them or even increase the landscapes' potential to supply ES without producing tradeoffs. Potential ES supply is the maximum biophysically possible supply of a given ES in the absence of societal demand for, or benefits derived from, such services [39,40]. Coastal landscape boundaries are defined as the areas between 50 m below mean sea level and 50 m above the high tide level, or extending landward to a distance 100 km from shore [2].

To fill the aforementioned knowledge gap, this study investigates how LULCC in the coastal landscapes influence the quantities of provisioning and regulating ES supply and their spatial and temporal distribution over the landscape. It also explores how social and environmental drivers of LULCC affect ES supply potentials of the landscape. The implications of changes in ES supply potentials for land use planning in the region are discussed.

## 2. Materials and Methods

### 2.1. Study Area

This study was carried out in the Greater Amanzule Landscape located in Southwestern Ghana. This landscape falls within Ghana's Wet Evergreen Forest zone, which lies in the Upper Guinean Forest Ecosystem of West Africa. Covering approximately 60,000 ha, the landscape extends from the Ankobra River estuary, stretching to the Tano basin on Ghana's southwestern boundary with Cote d'Ivoire (Figure 1) [41]. The area is characterized by a bimodal rainfall regime, with peak rainfall occurring in May to June and October to November each year. Mean annual rainfall is 1600 mm with a relative humidity of 87.5% [42]. It encompasses a relatively pristine and vast expanse of coastal ecosystems comprising swamp forests, freshwater lagoons, rivers, mangrove forests, terrestrial forests, agricultural lands and grasslands. It is associated with a relatively high diversity of flora and fauna (237 species of plants, 27 species of mammals and 26 species of demersal fish) and known to be inhabited by most of Ghana's forest primate species [42,43]. The landscape traverses three district boundaries. It is a community-protected area and awaits official government designation as a conservation area. Farming is largely subsistence and a source of nutrition for the growing population. Increasingly, plantation agriculture, notably rubber and oil palm, are becoming attractive land use options for land owners and the agro-based private sector. Culturally, mangrove wood is the preferred fuelwood for smoking fish in traditional ovens [42]. The discovery of oil and gas in commercial quantities off the continental shelf in Southwestern Ghana ushered the region into a new wave of competition between industrial, residential and agricultural land uses [36]. This is manifested by the losses of farmland and forests in favor of built-up areas in the region's urban core and peripheries [36,44]. It is noteworthy that onshore oil and gas infrastructure is expanding into ecologically sensitive areas of this landscape, thereby causing further habitat fragmentation and threatening wildlife [45]. The population has doubled over the last decade and is increasing above the national average due to the region being a focal point for in-migration [44]. Historically, economic development in this area was driven by a vibrant fishing industry, but more recently the fisheries sector has suffered decline [46]. Similar to other coastal regions in

Ghana, the well-being of the local population is inexorably linked to natural resources which underpin their contentment with ES [47].

**Figure 1.** Map of the coastal landscapes of Southwestern Ghana showing the study area.

### 2.2. Methodological Framework

A stepwise and iterative process was utilized, as illustrated in the methodological framework to assess land use impacts on coastal ES (Figure 2). In the first step, we identified and selected relevant provisioning and regulating ES on the basis of available data and relevance to spatial planning in Southwestern Ghana. In the second step, we conducted land cover classification for the study landscape through application of remote sensing techniques. Thirdly, land use/land cover types were matched with ES, and finally, using benefit transfer approaches, the landscape's potential to supply provisioning and regulating ES was quantified. Results of the ES supply potential for the study landscape were spatially represented using 2000 and 2018 as temporal reference points.

### 2.3. Data Types and Sources

To enable assessment of land use impacts on ES supply potentials, a representative landscape of 362 km² was delineated on the basis of the following criteria: (a) representativeness of regional ecological (critical watersheds) and sociocultural characteristics (different land use intensities), and (b) availability of cloud-free satellite images for the assessment timeframe. We used two temporal reference points to depict important milestones in regional land uses in the study area, which in turn provided the basis for comparing land use impacts on the landscape potential to supply ES over time.

The study utilized two main remote sensing datasets (Landsat Thematic Mapper (TM) and Landsat Operational Land Imager (OLI)), which were acquired from the United States Geological Survey (USGS) web data repository. Landsat OLI was acquired for December 2016 and January 2018 and combined into a single image. The Landsat TM was acquired for February 2000 and January 2002. All images from Landsat sources had 30 m spatial resolution. Orthorectified images (at 5 m spatial resolution) were acquired for 2005 from the Ghana Geological Survey. Google Earth images were used for data verification. The study also relied on nonspatial data collected from published sources, gray literature and agricultural statistics. Assessment of the landscapes' potential to supply food-crop

provisioning services was based on agricultural data collated by the Ministry of Food and Agriculture (MOFA) at the regional level. For estimating the landscapes' potential to supply fuelwood and sequester carbon, data from ecological surveys conducted in the study region and comparable ecosystems along Ghana's coast were utilized.

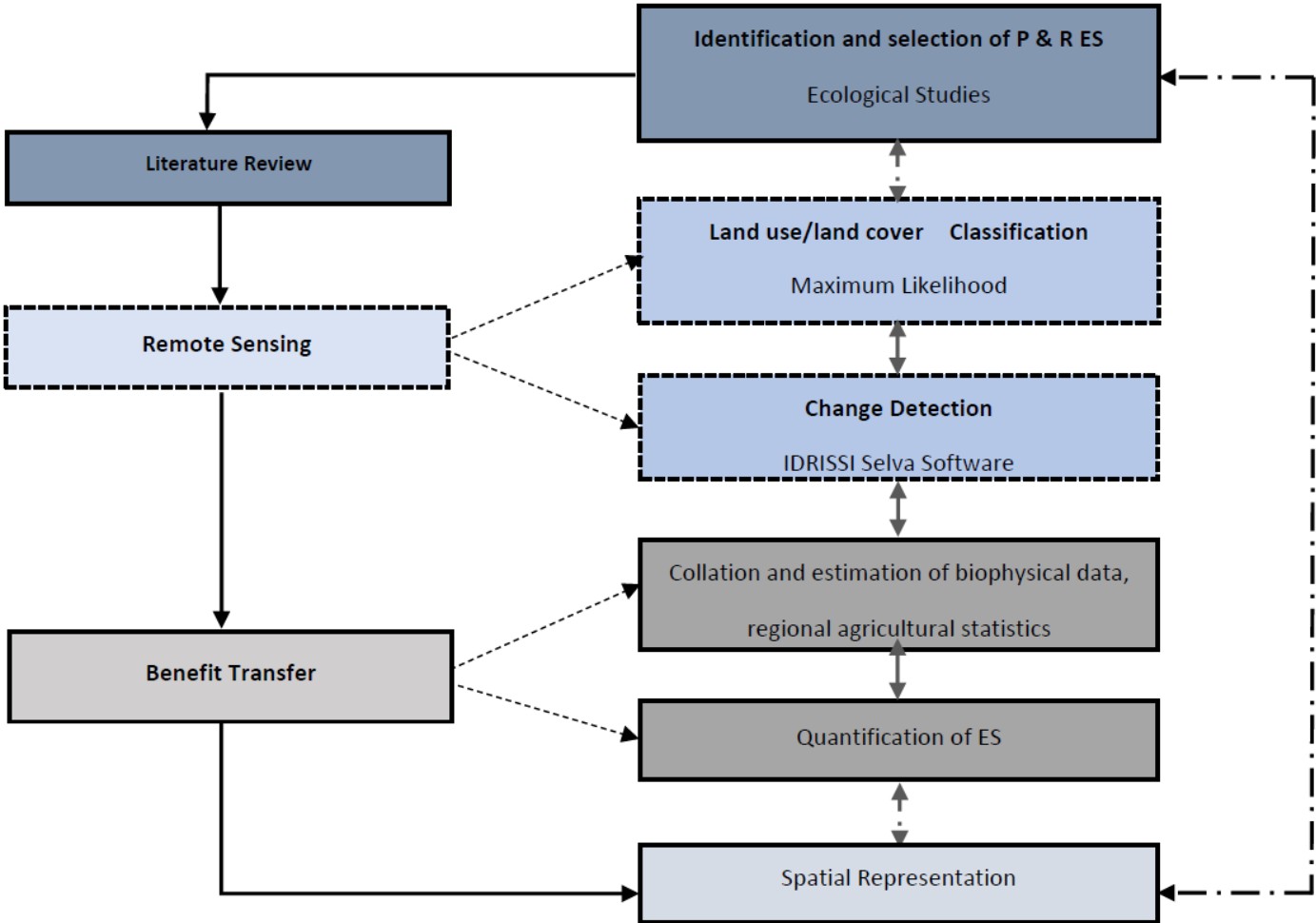

**Figure 2.** Methodological framework for assessing land use impacts on coastal ES in Southwestern Ghana.

### 2.3.1. Remote Sensing Data Processing and Analysis

Landsat 5 TM (Thematic Mapper) data from 2000/2002 and Landsat 8 OLI (Operational Land Imager) data from 2016/2018 provided by the United States Geological Survey (USGS) Earth Explorer database system were used for generating land use/land cover maps. All of the raw images were taken in the same season and nearly free of clouds. For each of these periods, tiles from 2 dates were selected and combined to provide a single image for the study area. All the processing and post-classification steps were completed using the software packages Erdas Imagine 2015 and ArcGIS 17.1. Prior to interpretation, image pre-processing including geometric and radiometric corrections was performed for each of the images. All of the data were geometrically corrected and projected to Universal Transverse Mercator (UTM) zone 30 N. After image pre-processing, supervised classification methods and maximum likelihood algorithms were used for preparing land use/land cover maps for two temporal reference points. The land use/land cover categories were assigned according to the descriptions of the Land Cover Classification System (LCCS), which is a hierarchical a priori classification scheme providing a flexible framework for identifying land use classes in highly heterogeneous landscapes such as those found in the study region [48]. Ten land cover classes were derived to match available data for ES quantification. Change analysis was conducted using the Land Change Modeler embedded in the IDRISI TerrSet software.

2.3.2. Selection of Provisioning and Regulating ES

Ecosystem services were selected from the list of land-cover-based proxies compiled in the literature and for mapping ES [17,49]. The types and sources of data and corresponding proxy indicators for ES quantification are presented in Table 1. The derived land cover classes (see Section 3.1) were the basis for representing ES supply with land cover types occurring in the study region [17]. Our proxy measures for carbon storage were based on data from primary ecological studies, which estimated aboveground carbon in multiple mangrove stands along Ghana's coast and also in rubber plantations [50–52]. Similarly, primary ecological studies that estimated aboveground tree biomass across mangrove stands were utilized as proxies for fuelwood supply. Food supply was based on land cover data combined with official agricultural statistics. Mangrove fuelwood was selected as it is a dominant source of fuelwood utilized in the local fishing industry, while plantain and cassava are the major staple food supply from cropland in the region. Additionally, mangroves store large quantities of carbon, and are increasingly receiving attention currently in Ghana's climate change mitigation strategies. In addition to latex production, rubber plantation development is arguably presented as a significant opportunity requiring inclusion in Ghana's climate change mitigation programs [53].

**Table 1.** Types and sources of data utilized for assessing ES in the coastal landscapes of Southwestern Ghana. LULC = land use/land cover types; USGS = United States Geological Survey; TM = Thematic Mapper; OLI = Operational Land Imager; MoFA = Ministry of Food and Agriculture; ES = ecosystem services; P = provisioning services; R = regulating services; Mg $C_{org}$ = quantities of organic carbon stored in vegetation; - = not applicable.

| Type of Data | Period | Sources of Data | Relevant LULC Types/ES | Proxy Indicator | Unit | References |
|---|---|---|---|---|---|---|
| Remote sensing | February 2000/2002 | USGS Landsat TM/Landsat OLI | Mangrove, Rubber, Cropland | - | - | https://earthexplorer.usgs.gov/ (accessed on 12 August 2022) |
|  | December 2016; January 2018 |  |  |  |  |  |
| Annual cassava and plantain yield | 2000–2016 | MoFA regional agricultural statistics | Food—P | Total crop yield | Tons | - |
| Mangrove forest stand biomass | 2015, 2016 | Ecological survey | Fuelwood—P | Total growing stock | Tons | [50,51,54] |
|  |  | Ecological survey | Carbon storage—R | Carbon stored in aboveground vegetation | Mg $C_{org}$ | [50,51,54] |
| Aboveground carbon in rubber tree stands | 2017 | Ecological survey | Carbon storage—R | Carbon stored in aboveground vegetation | Mg $C_{org}$ | [52] |

*2.4. Benefit Transfer*

Benefit transfer involves extrapolation of either biophysical measures or economic estimates from a previously estimated site to a study area of interest [27,28]. This is based on the assumption that spatial units are homogenous; hence, estimates from one area is transferable to the other [31]. We compiled biophysical values from ecological studies conducted in mangrove forests and rubber plantations along with crop yield estimates from regional agricultural statistics (Supplementary Table S1). Extrapolated mean values from the ecological studies and agricultural statistics were assigned to the corresponding land use/land cover types in GIS. This ensured that generalization errors were minimized and better correspondence was achieved in the biophysical characteristics between the previously estimated sites and the study landscape [55]. Using the image resampling tool in ArcGIS Pro, we resampled the 30 m × 30 m land cover data to hectares. In estimating the landscapes' potential to supply ES, we multiplied the mean values computed from the ecological studies and agricultural statistics by the resampled land cover data.

2.4.1. Quantification of Provisioning Services Supply Potentials

Mangrove Fuelwood

Mangrove forests are adapted to tropical and subtropical coastal environments [38,39]. There is widespread harvesting and utilization of mangroves as sources of fuelwood for fish smoking in the coastal areas of Ghana. Mangrove fuelwood is hereby defined as wood harvested from live trees and standing dead wood. The dominant mangrove species found in the study region are *Rhizophora mangle*, *Avicennia germinans* and *Laguncularia racemosa* [50,51]. Adotey [50] and Nortey et al. [51] utilized allometric equations derived from diameter at breast height (DBH) and height (H) to estimate aboveground (standing dead wood and live trees) biomass of mangroves found along four river estuaries (*Whin*, *Amanzule*, *Kakum* and *Nyan)* located on Ghana's western and central coasts. In this study, we estimated the potential of the landscape to supply mangrove fuelwood by calculating the mean aboveground biomass across all the four sites sampled by Adotey [50] and Nortey et al. [51] according to the formula:

$$M_{ABG} = B_{S1} + B_{S2} + B_{S3} + B_{S4}/Ns, \qquad (1)$$

where $M_{ABG}$ = mean aboveground biomass, $B_{S1}$ = biomass at site 1, $B_{S2}$ = biomass at site 2, biomass at site 3, $B_{S3}$ = biomass at site 4 and $N_s$ = number of sites. The mangrove fuelwood supply potential of the landscape was mapped in GIS and the results compared over the two temporal reference points.

Food Production

Staple food crops in Southwestern Ghana comprise cassava, yam, cocoyam, rice, maize and plantain. At the regional level, the Ministry of Food and Agriculture (MOFA) maintains a database of crop production, crop yield and area cultivated. Cassava and plantain comprise over 80% of total food-crop production in the region. In assessing the food-crop supply potential of the landscape, we extracted the staple crop yield statistics of the three districts—Ellembelle, Nzema East and Jomoro—that span the study landscape. The mean yield of cassava and plantain was estimated for the period 2000 to 2018 according to the formula:

$$MC_{yield} = [Y_1 + \ldots \ldots Y_n]/N_y prod, \qquad (2)$$

$$MPl_{yield} = [Y_1 + \ldots \ldots Y_n]/N_y prod, \qquad (3)$$

where $MC_{yield}$ = mean cassava crop yield in tons$^{-ha}$, $Y_1$ = yield in the first year of production in tons$^{-ha}$, $Y_n$ = yield in last year of production in tons$^{-ha}$, $MPl_{yield}$ = mean plantain crop yield in tons$^{-ha}$ and $N_y$ = number of years of staple food-crop production.

The estimated mean yield of the major staple crops was multiplied by the area of rainfed cropland in the land use/land cover map to estimate the potential food supply in tons, as per the formula:

$$P_{fs} = M_{yield} \times A_{cropland} \qquad (4)$$

where $P_{fs}$ = potential food-crop supply and $A_{cropland}$ = area of cropland. The food-crop production potential of the landscape was mapped using GIS and the results compared over the two temporal reference points.

2.4.2. Quantification of Regulating Services Supply Potentials

Mangrove Carbon Storage

Mangrove ecosystems are globally recognized for their significant contribution to carbon cycling and sequestration [56–58]. Mangrove ecosystem carbon pools are stored in aboveground biomass, belowground biomass, litter and soil organic matter components [43,45]. While protection of mangroves contributes to attainment of climate change mitigation objectives, their conversion to other land cover types is a significant source of carbon emissions into the atmosphere. Carbon quantity stored in mangroves is estimated using allometric equations that

relate biomass with parameters such as diameter at breast height, height and density of mangrove trees [59]. Adotey [50] and Nortey et al. [51] utilized species and site-specific allometric equations to quantify mangrove carbon stocks across sample plots established in mangrove ecosystems found in the Amanzule, Kakum, Nyan and Whin river estuaries located along the coast of Ghana. In this study, we derived proxy data from Adotey [50] and Nortey et al. [51] to quantify aboveground carbon stored in mangrove ecosystems found in the study landscape. We narrowed and focused on aboveground carbon pools as these are better reflected in the mangrove vegetation captured using satellite data. Per hectare aboveground mangrove biomass estimates from the abovementioned sample sites were summed and multiplied by a conversion factor [50] and the average determined according to the formula:

$$M_{AGC} = [B_{S1} + B_{S2} + B_{S3} + B_{S4}] \times 0.46/N_s, \tag{5}$$

where $M_{AGC}$ = Mean aboveground carbon; $B_{S1}$ = biomass at site 1; $B_{S2}$ = biomass at site 2; $B_{S3}$ = biomass at site 3; $B_{S4}$ = biomass at site 4; $N_s$ = number of sites; 0.46 = conversion factor for tropical mangroves. The mangrove carbon storage potential of the landscape was quantified by multiplying the estimated mean aboveground carbon stored in mangroves ($M_{AGC}$) by the mangrove extent in the land use/land cover map using the formula;

$$P_{cs} = M_{AGC} \times A_{mangrove} \tag{6}$$

where $P_{cs}$ = landscape potential to store carbon and $A_{mangrove}$ = area of mangrove. The mangrove carbon storage potential of the landscape was mapped using GIS and the results compared over the two temporal reference points.

Rubber Carbon Storage

Growing market demand for natural rubber on the international market is partly driving expansion of rubber plantations in the tropics [60]. Because of land use competition between rubber plantation and tropical forestry, the potential role of rubber plantations in ecosystem services provisioning is gaining scholarly attention (e.g., [46,47]). Carbon in a rubber plantation is stored in aboveground and belowground biomass and in latex and soil [52]. The quantity of carbon stored in rubber varies with the age of trees in a plantation. Site specific allometric equations have been developed for estimating carbon sequestered in rubber plantations [53]. Using data collected from 25 sample plots in rubber plantations located in Ghana's western region, Tawiah et al. [52] integrated data on the age of rubber plantations, field-measured diameter at breast height and latex production in allometric equations to estimate aboveground, belowground and latex carbon. In this study, we utilized proxy values from Tawiah et al. [52] to estimate aboveground carbon for the study landscape according to the formula:

$$M_{AGC} = M_F + M_S + M_L \tag{7}$$

where $M_{AGC}$ = mean aboveground carbon, $M_F$ = mean foliage carbon, $M_S$ = mean stem carbon and $M_L$ = mean latex carbon. Potential carbon storage in rubber plantation was estimated by multiplying the estimated mean aboveground carbon by the extent of rubber in the land use/land cover map using the formula;

$$P_{cs} = M_{AGC} \times A_{rubber} \tag{8}$$

where $P_{cs}$ = landscape potential to store rubber carbon and $A_{rubber}$ = area of rubber.

*2.5. Social and Environmental Drivers of LULCC*

We utilized the Exploratory Regression tool in Arc GIS Pro to explore the relationships between pre-selected independent variables (elevation, slope, rainfall, fishing population, farming population, community population, distance from gas pipeline, distance from oil processing plant, distance from rubber processing facility, distance from river and distance

from sea) and the dependent variables (land use/land cover transitions) in the study region. Pre-selection of independent variables was informed by literature on driving forces of LULCC in tropical regions [61]. The dependent variables were conversions to cropland, rubber plantations and mangroves since they constitute land use/land cover classes for the supply of relevant ES. Rapid landscape transformations emanating from industrial activities provided an additional basis for exploring the effects of the independent variables on conversions to artificial/bare areas [34].

Subsequently, we ran geographically weighted regression (GWR) to predict the effect of changes in the high-performing independent variables on land use/land cover outcomes. The model parameters for transitions to artificial/bare areas were elevation, slope, distance from road, fishing population, farming population and total resident population. Parameters for transitions to rubber plantation were elevation, slope, distance from road, total resident population and farming population. Parameters for transitions to cropland were elevation, slope and distance from road. The aim of the GWR was to explain the spatial variations in the relationships between the independent variables and the dependent land use/land cover transitions.

## 3. Results

### 3.1. Land Use/Land Cover Changes in 2000 and 2018

The land use/land cover maps shown in Figure 3 and the extent of changes in the land use/land cover classes depicted in Table 2 were generated as inputs for assessment of the selected ES. Next to wetlands, cropland dominated the land use/land cover situation in the study area, representing approximately 24 and 35% coverage in 2000 and 2018, respectively. Within the cropland category are cassava and plantain, which are the major staple crops in the region and "others" subclasses. Cassava dominated the cropland category, increasing from 10 to 15% of total land use/land cover in 2000 and 2018, respectively. This was followed by the "others" subclass, which occupied 9% of the area in 2000 and increased to 12% in 2018. Plantain occupied 5% and increased to 8% of total land use/land cover in 2000 and 2018, respectively. Grassland decreased markedly, from 11 to 4%, and shrubland/sparse vegetation reduced from 19 to 8% in 2000 and 2018, respectively. Artificial/bare areas and rubber plantation showed sharp increases from 7 to 11% and from 1 to 5%, respectively, in 2000 and 2018. However, mangrove cover remained relatively stable at approximately 1% in 2000 and 2018.

### 3.2. Quantities, Spatial and Temporal Distribution of ES Supply Potentials

#### 3.2.1. Mangrove Fuelwood Supply

The landscape's potential to supply mangrove fuelwood ranged from a minimum of 0.01 tons to a maximum of 87.19 tons, as shown in Figure 4. For the two temporal reference points, marked differences in the spatial distribution of mangrove fuelwood supply potential were also observed across the landscape (Figure 4). The potential mangrove fuelwood supply was concentrated along the intertidal areas extending eastward from Sanzule and Essiama. Similarly, between 2000 and 2018, mangrove fuelwood supply potential was relatively high on the coastlines stretching eastward from Sanzule, while the supply remained relatively stable over these two temporal horizons for the same location, as depicted in Figure 4A. Comparatively, mangrove fuelwood supply potential sharply decreased along the coastal stretches westward and eastward from Essiama, as shown in Figure 4C. At these locations, the decrease in mangrove fuelwood supply potential was even more pronounced between 2000 and 2018, as illustrated by Figure 4B,C.

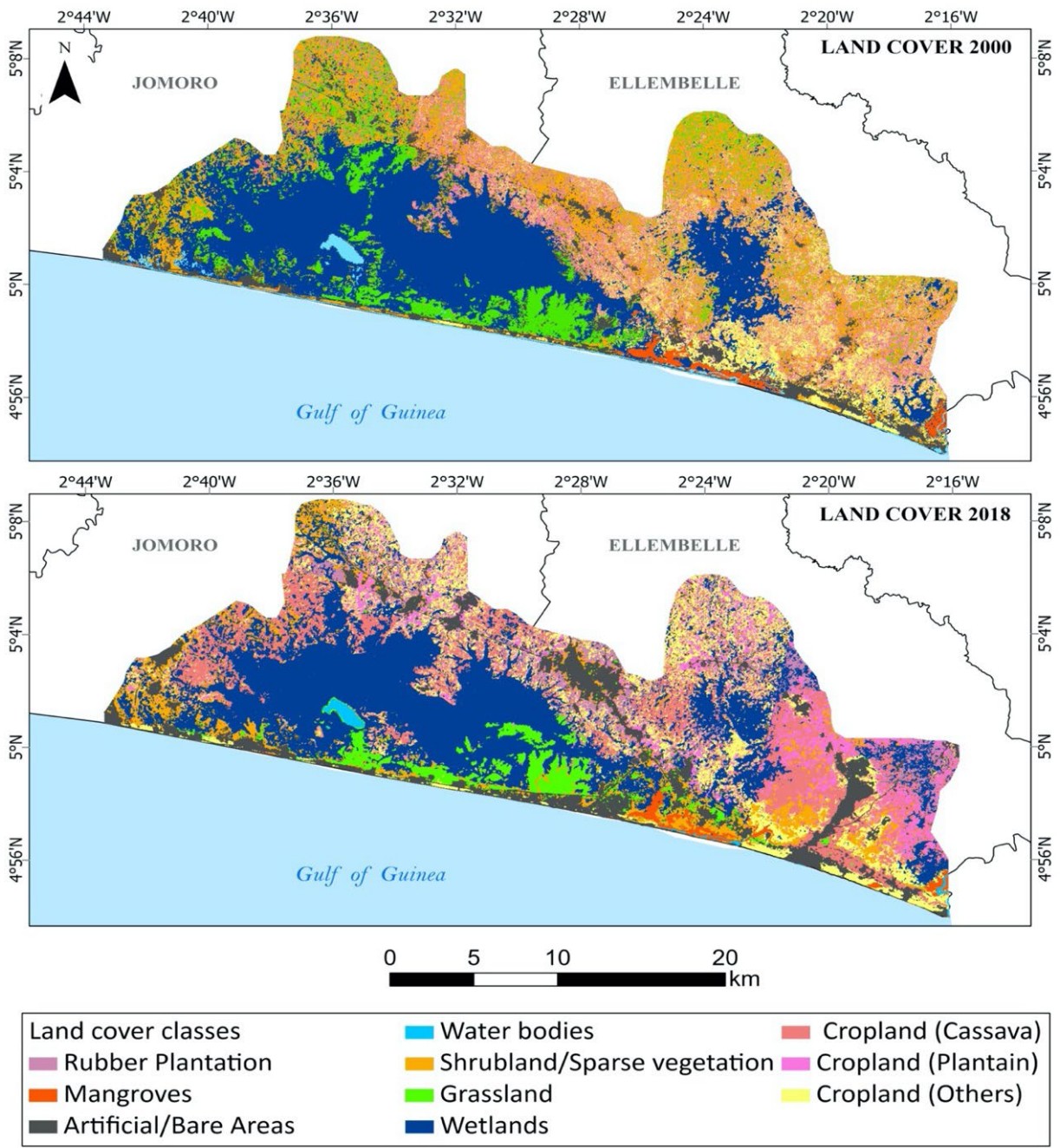

**Figure 3.** Land use/land cover types in the study area.

**Table 2.** Main land use/land cover types, their extent and description.

| LULC Types | Extent (%) | | Description |
|---|---|---|---|
| | 2000 | 2018 | |
| Mangroves | 1.18 | 1.02 | Coastal forests of stilted shrubs or trees bordering the ocean or coastal estuaries, composed of one or several mangrove species. |
| Wetlands | 34.92 | 34.15 | Herbaceous or aquatic vegetation in permanent or semipermanent swamps. |
| Rubber plantation | 1.49 | 5.38 | Regular stands of trees planted for the purpose of producing materials for industry. |

**Table 2.** *Cont.*

| LULC Types | Extent (%) | | Description |
|---|---|---|---|
| Artificial areas/bare areas | 7.02 | 11.27 | Cover resulting from human activities such as urban development, extraction or deposition of materials. It comprises areas that are not covered by vegetation, such as rocky or sandy areas. |
| Grassland | 10.71 | 4.04 | Mixed mapping unit that consists of 50–70% grassland. |
| Shrubland/sparse vegetation | 19.38 | 8.34 | A class representing a mapping unit which contains 20–10% to 1% vegetative cover. |
| Water bodies | 1.39 | 0.46 | Areas covered by natural water bodies such as ocean, lakes, ponds, rivers or streams. |
| Cropland_Cassava | 9.55 | 15.44 | Mix crops and nonforest vegetation with cassava representing more than 90% of the cover. |
| Cropland_Plantain | 4.94 | 7.62 | Mix crops and nonforest vegetation with plantain representing more than 90% of the cover. |
| Cropland_Others | 9.40 | 12.27 | Mix crops and nonforest vegetation with croplands representing more than 50% of the cover. |

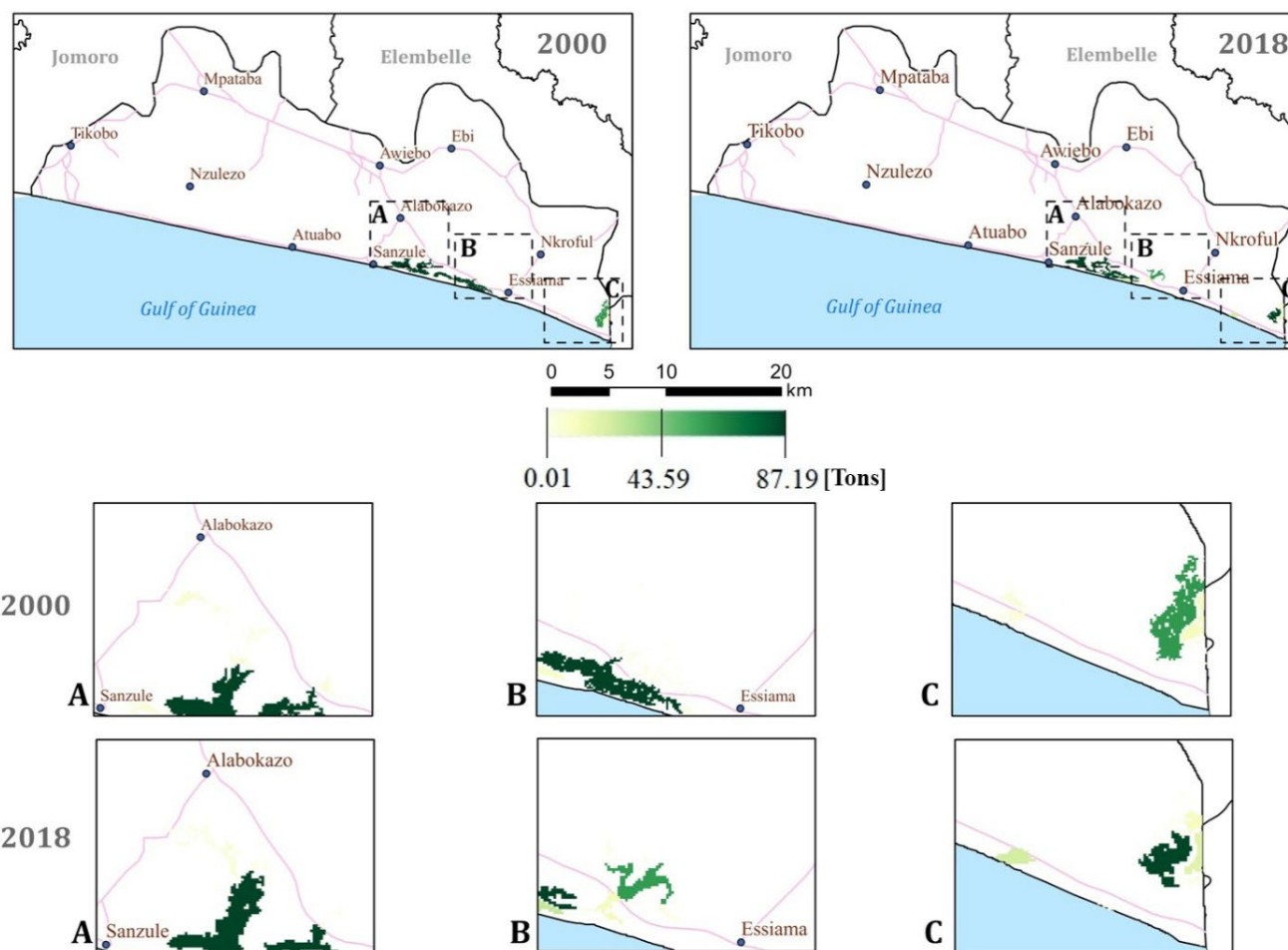

**Figure 4.** Quantities of mangrove fuelwood supply and distribution across the landscape in 2000 and 2018; (**A**–**C**) compares the spatial and temporal distribution of mangrove fuelwood supply potential in segments of the landscape.

### 3.2.2. Food-Crop Supply

Cassava Supply

The minimum and maximum potentials of the landscape to supply cassava food crop ranged from 1.7 to 23.4 Gt, as shown in Figure 5. Generally, the spatial distribution patterns of potential cassava food-crop supply showed skewness toward the southeastern, northeastern and northwestern portions of the landscape. Comparison of cassava food-crop supply potential between the two temporal reference points also showed higher potential supply in 2018 than 2000, as depicted in Figure 5A–D. Furthermore, during 2018, cassava supply potential was more spatially concentrated within the northern sections of the landscape and in areas within close proximity to road networks and major towns such as Tikobo, Nkroful, Alabokazo and Essiama.

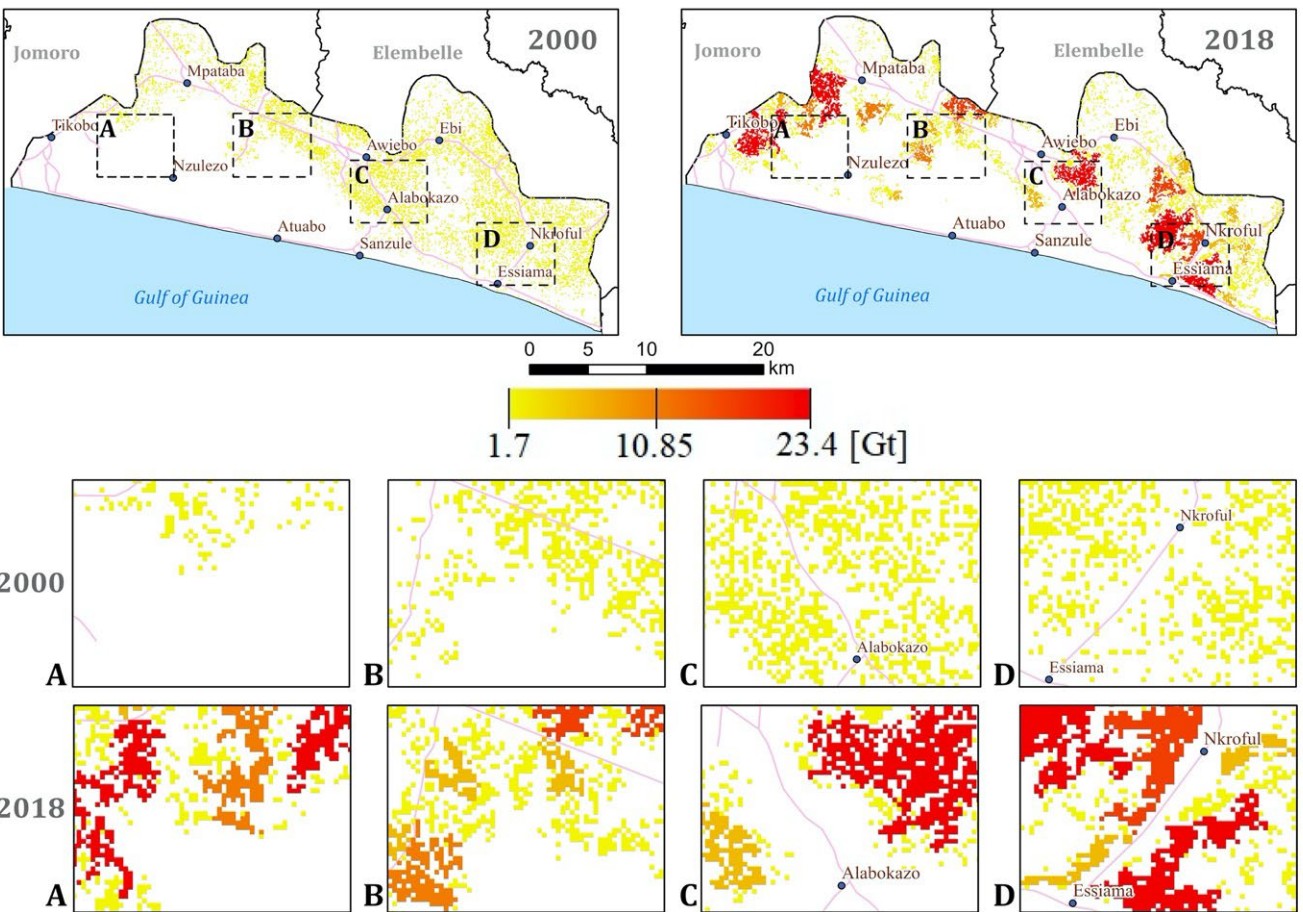

**Figure 5.** Quantities and distribution of food supply from cassava across the landscape in 2000 and 2018; (**A–D**) compares the spatial and temporal distribution of cassava food-crop supply potential for segments of the landscape.

Plantain Supply

The potential of the landscape to supply plantain food crop is depicted in Figure 6. While the minimum and maximum potential supply ranged from 0.001 to 3.3 Gt for the two temporal reference points, potential supply was found to be higher during 2018 than 2000. However, as shown in Figure 6D, the distribution patterns of potential plantain supply in Nkroful and Essiama was less in 2018 compared to 2000.

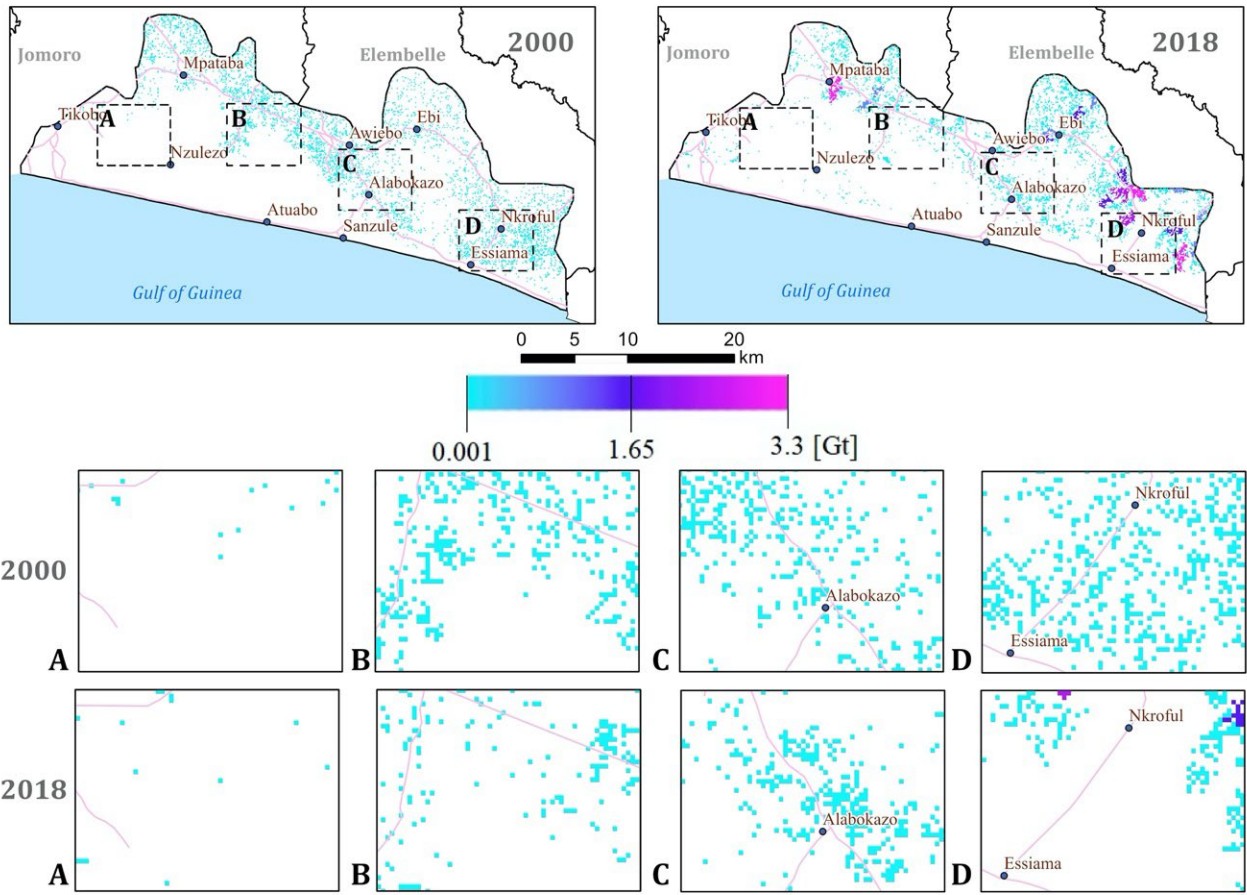

**Figure 6.** Quantities and distribution of food supply from plantain across the landscape in 2000 and 2018; (**A**–**D**) compares the spatial and temporal distribution of plantain food-crop supply potential for segments of the landscape.

### 3.2.3. Carbon Storage
Mangrove Carbon

The aboveground carbon storage potential of mangrove forests in the study landscape ranged from a minimum of 6.6 Mg C to a maximum of 87,196 Mg C, as shown in Figure 7. Spatial and temporal patterns of mangrove carbon storage potential remained relatively unchanged across the landscapes. However, the spatial distribution pattern of aboveground mangrove carbon storage potential showed variations along the coast. Potential for mangrove carbon storage was concentrated in the southeastern portions of the landscape: along the coastal stretch between Sanzule and Essiama, eastward from Essiama and along the lower Ankobra riparian areas (Figure 7A–C), whereas the landscape showed no potential for mangrove carbon storage along the southwestern side of the coast.

Rubber Carbon

The aboveground rubber carbon storage potential of the landscape is shown in Figure 8. This ranged from a minimum of 0.05 Mg C to a maximum of 429 Mg C. Nonetheless, as indicated in Figure 8, rubber carbon storage potential skewed toward the lower limit, between 0.05 to 257.40 Mg C. The distribution pattern of rubber carbon storage potential also showed clustering on the central and northern portions of the landscape. Additionally, this potential increased markedly in 2018 compared to the situation in 2000 (Figure 8A–D).

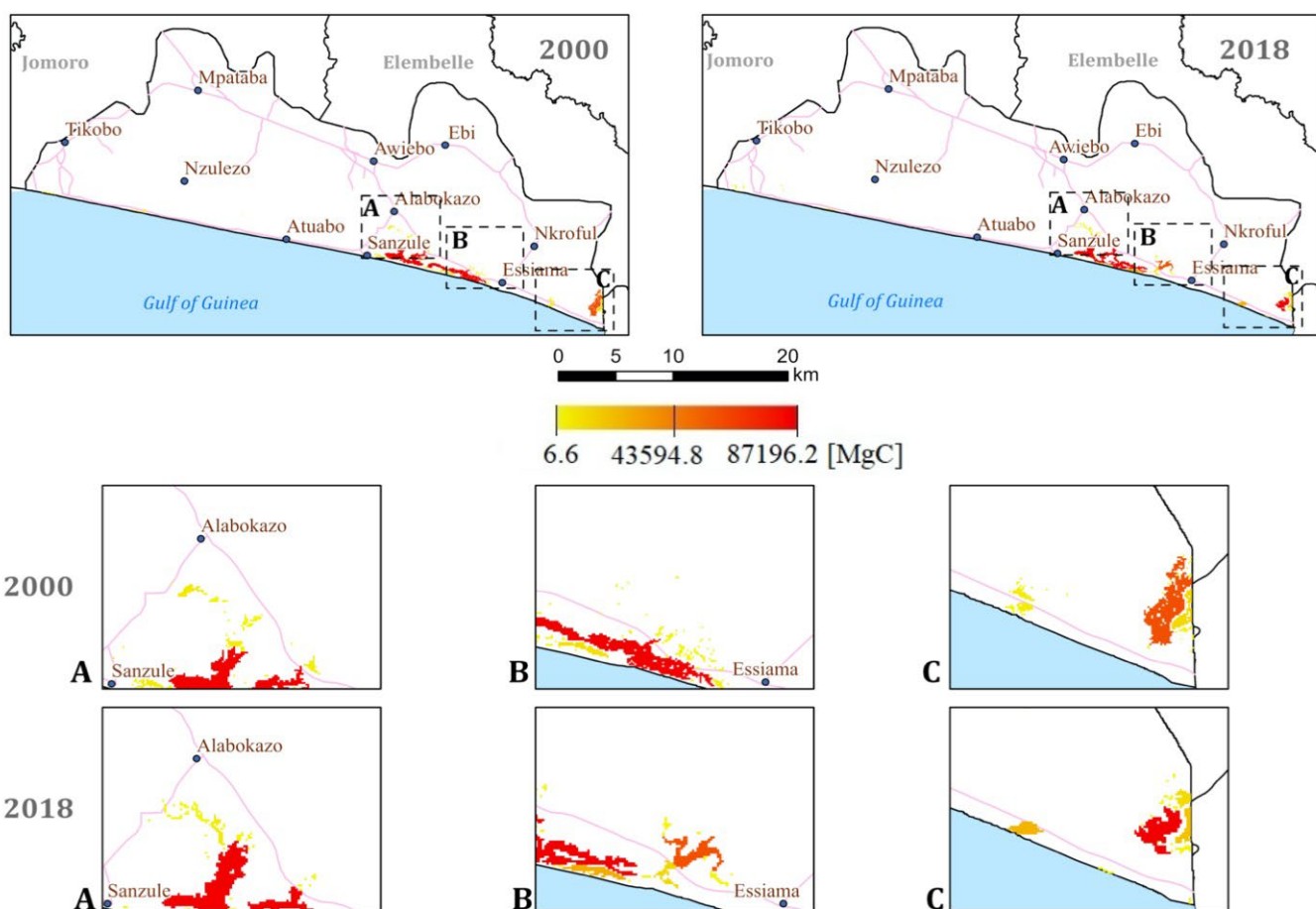

**Figure 7.** Quantities and distribution of aboveground carbon stored in mangrove forest across the landscape in 2000 and 2018; (**A**–**C**) compares the spatial and temporal distribution of aboveground carbon storage potential of sections of the landscape.

*3.3. Predictors of LULCC*

Geographically weighted regression models showing predictors of transitions to artificial/bare areas, rubber plantation and cropland are depicted in Figures 9–11, respectively. The predictors—farming population, total resident population, elevation and slope—exhibited spatial variations in their relationships with transitions to artificial/bare areas. Within the major towns such as Atuabo, Beyin and Krisan, there were strong spatial effects between farming population ($\beta$ = 3.09–16.31, $R^2$ = 0.6–0.9) and transitions to artificial/bare areas. Similarly, as shown in Figure 9, resident population and slope positively affected transitions to artificial/bare areas. These effects were strong in Awiebo and surrounding towns. On the other hand, elevation, total resident population and farming population showed strong effects with land use/land cover transitions to rubber plantation. These effects were exhibited at the northwestern and northeastern portions of the landscape (Figure 10). Elevation exhibited strong effects with transitions to cropland at Tikobo No. 1, Mpataba and Awiebo. Similarly, slope showed strong spatial effects with transitions to cropland within the foregoing locations and around Nkroful, Esiama and Asanta (Figure 11).

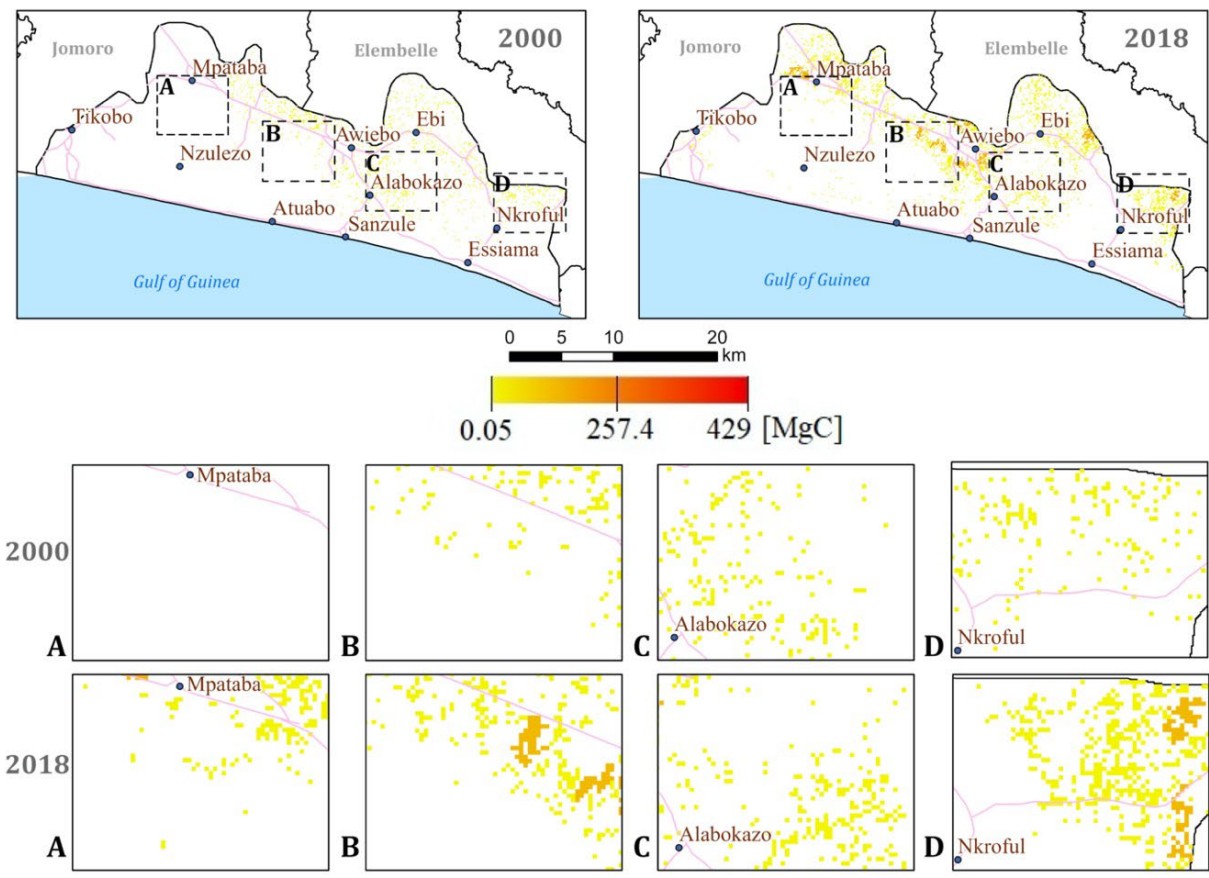

**Figure 8.** Quantities and distribution of aboveground carbon stored in rubber plantation across the landscape in 2000 and 2018; (**A–D**) compares the spatial and temporal distribution of aboveground rubber carbon storage potential of segments of the landscape.

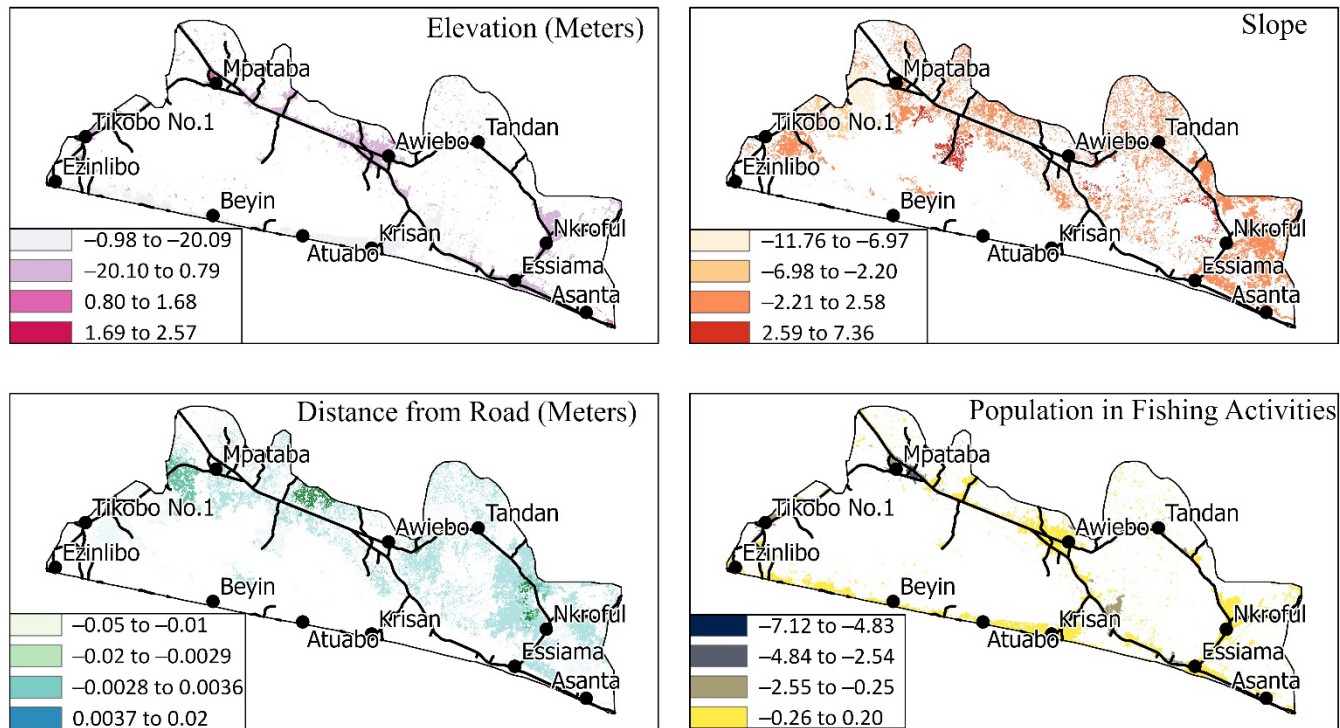

**Figure 9.** *Cont.*

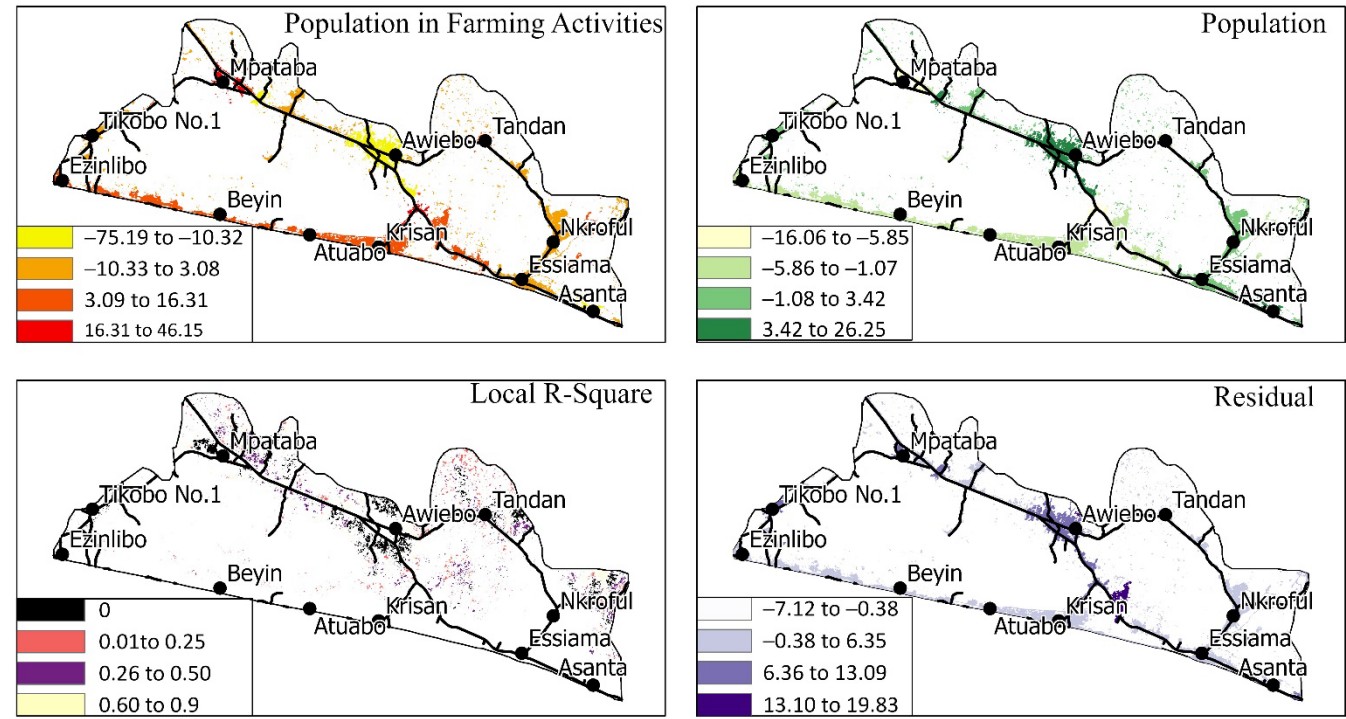

**Figure 9.** Results of geographically weighted regression showing spatial variations in the relationships between elevation, slope, distance from road, fishing population, farming population, total resident population and land use/land cover transitions to artificial/bare areas. Positive coefficients indicate the magnitude of spatial variation. Negative coefficients indicate no spatial correlations between the factor and transitions to artificial/bare areas.

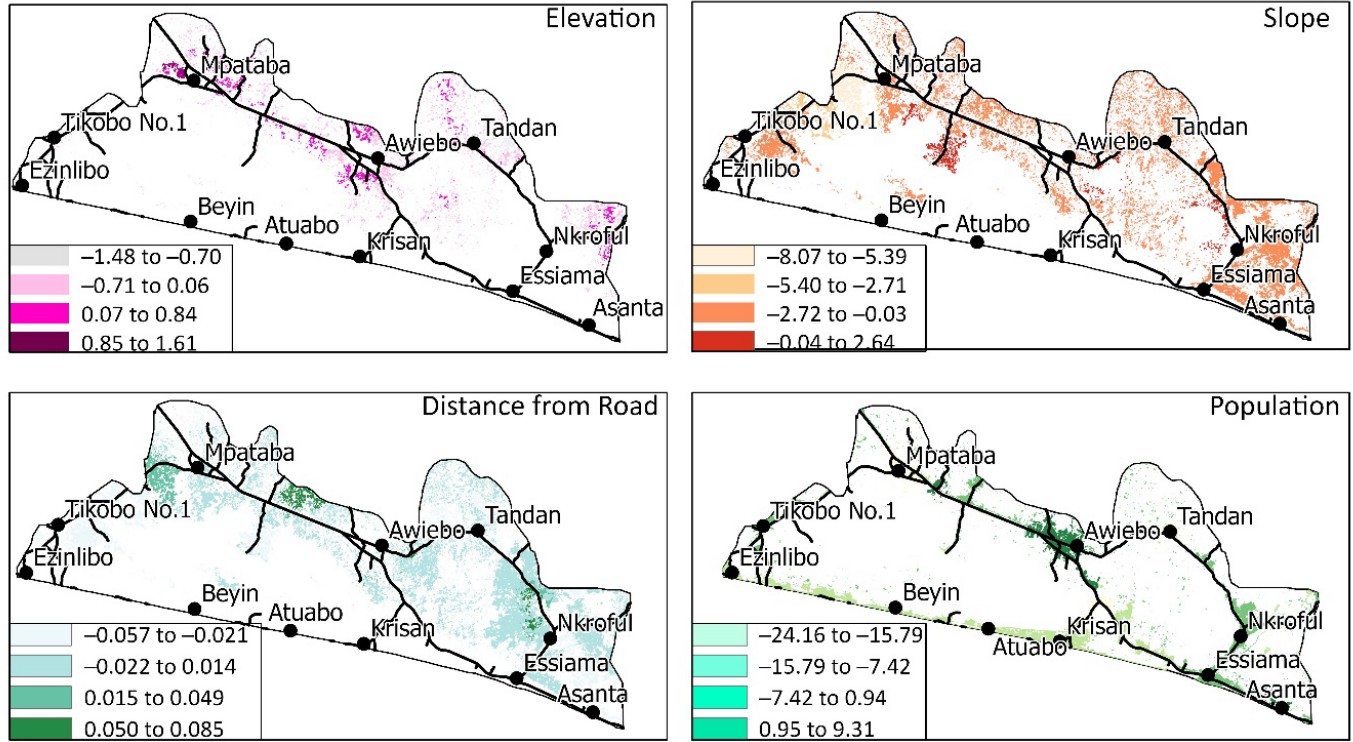

**Figure 10.** *Cont.*

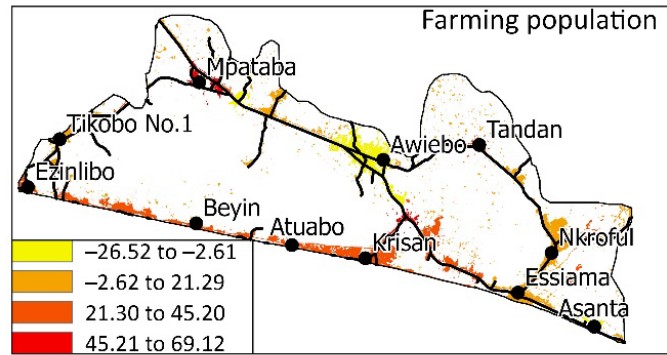

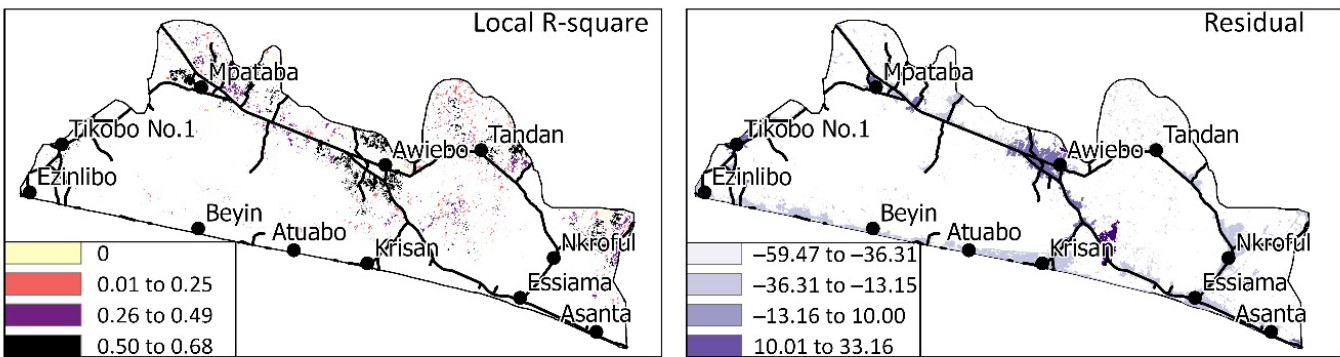

**Figure 10.** Results of geographically weighted regression showing spatial variations in the relationships between elevation, slope, distance from road, farming population, total resident population and land use/land cover transitions to rubber plantation. Positive coefficients indicate the magnitude of spatial variation. Negative coefficients indicate no spatial correlations between the factor and transitions to rubber plantation.

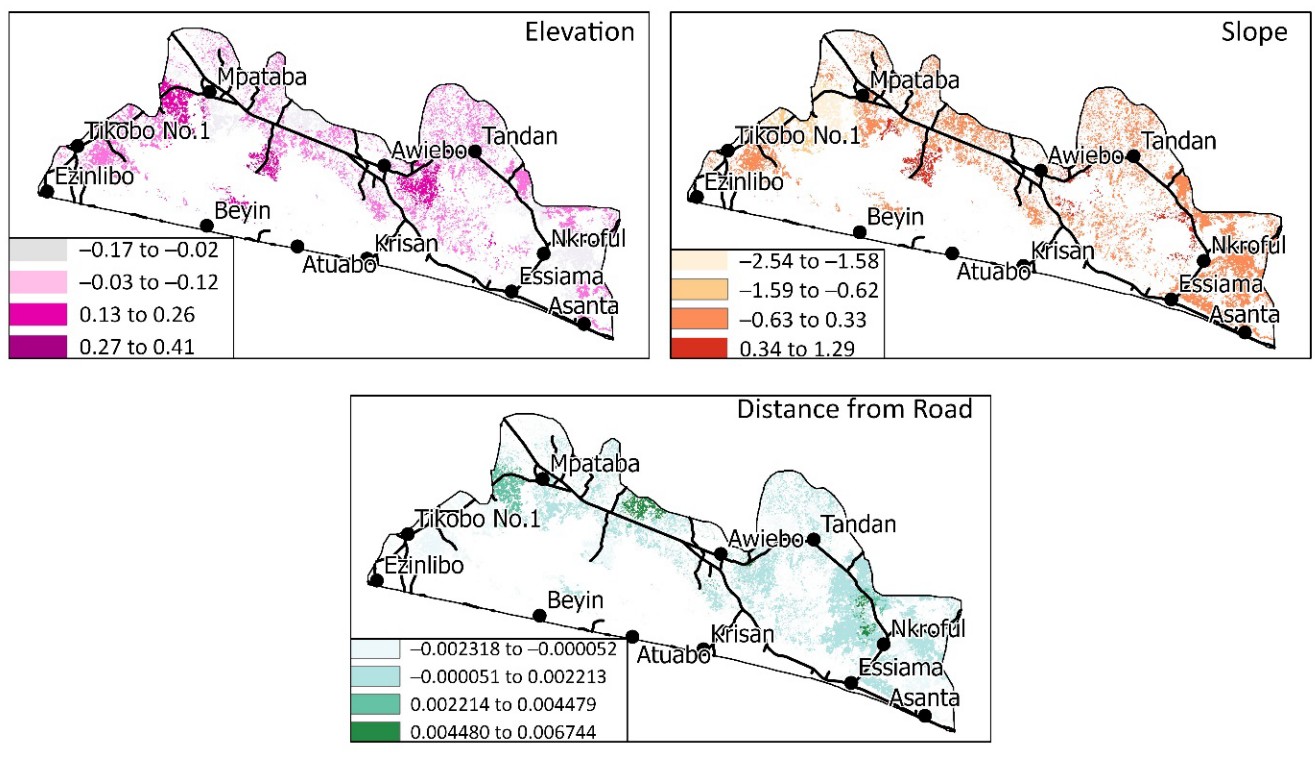

**Figure 11.** *Cont.*

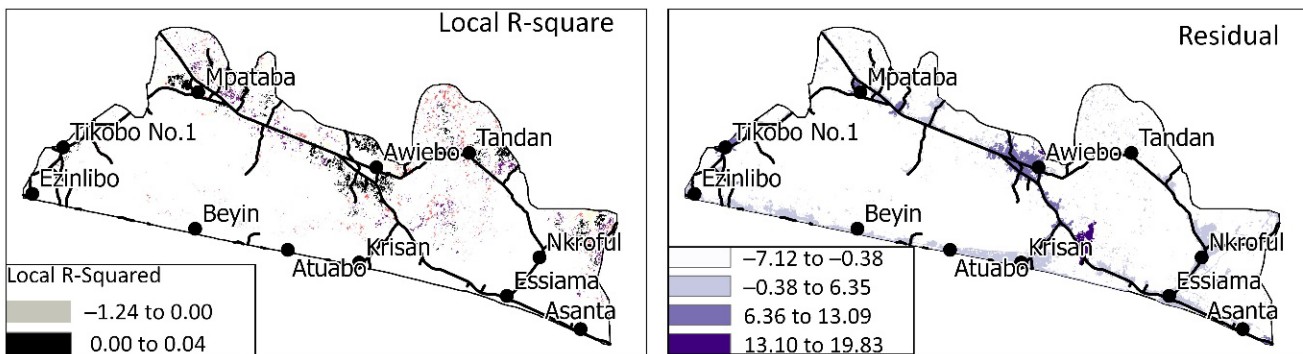

**Figure 11.** Results of geographically weighted regression showing spatial variations in the relationships between elevation, slope, distance from road and land use/land cover transitions to cropland. Positive coefficients indicate the magnitude of spatial variation. Negative coefficients indicate no spatial correlations between the factor and transitions to cropland.

## 4. Discussion

### 4.1. Land Use/Land Cover Changes and ES Supply Potentials

Coastal areas are rich in biodiversity and provide a variety of ES that are the basis for human well-being and sustainable development in most West African countries. The potential of coastal landscapes to sustainably supply provisioning services such as food and fuelwood are fundamental to livelihood improvement for the majority of coastal dwellers in the subregion. However, the growing conflicts between uses and among users of coastal resources require a better understanding of the impacts of human modification of coastal environments on the supply of ES [62]. In the coastal landscapes of Southwestern Ghana, land systems transformation is rapid and characterized by degradation of mangrove forests, rapid expansion of rubber plantation and infrastructure projects related to oil and gas activities [63,64]. Despite the potential negative impacts of such land conversions on ES, our findings point to increasing landscape potential to supply food. Increase in food production is inevitable, considering the food requirements of the growing population in the study region. Expansion of the area under cultivation and adoption of sustainable agricultural intensification practices while reducing land degradation are probable pathways for enhancing food production. Cropland constitutes the largest share of LULC type in the region, and this share has increased during the study period. Expansion in the area utilized for crop cultivation is a major contributing factor to the increase in food supply. This finding is congruent with similar studies that reported consistent increases in the value of ecosystem services for food production due to agricultural land expansion [13,65]. We narrowed and focused assessments to determine the relative potential to supply cassava and plantain, since they constitute the major staple crops in the study region. Our findings showed that food supply potential of the landscape was higher for cassava compared to plantain. Maximum cassava supply potential was seven times greater than the maximum plantain supply. The higher landscape potential to supply cassava is likely the result of expansion in the area of land under cassava cultivation between the different time periods. In the study region, land use for the purposes of agriculture is unregulated as the present land use system fails to allocate areas for specific crop types. Land allocation for arable crop production needs to consider biophysical constraints such as slope, elevation and soil suitability [66,67]. The absence of such land use guidelines increases the probability for conversion of fertile agricultural lands for food production in favor of industrial development, thereby risking food insecurity. Findings of a study conducted in adjacent coastal landscapes of Southwestern Ghana suggest that rapid spatial transformations characterized by rubber plantation and settlement expansion into traditional food-crop lands threatens local food production [34].

Mangroves provide wood and nonwood products, and are also critical for conservation of biological diversity while providing habitats and spawning grounds for a variety of fish

and shellfish [68]. Similar to many coastal areas in West Africa, coastal dwellers in the study region rely heavily on mangrove fuelwood for fish smoking [42,69]. However, we found that the landscapes' potential to supply mangrove fuelwood in the region is decreasing through time. Threats to mangrove forests vary across locations on the study landscape. For mangrove forest stands within proximity to small-scale fish markets, a supply chain for mangrove fuelwood is enabled by localized exploitation for fish processing. This supply chain is characterized by harvesting from the Ankobra riparian system, over relatively short distances by river and road transport to small scale fish processing destinations. Such mangrove fuelwood supply chains offer short-term economic returns to harvesters, thereby incentivizing further exploitation. Studies conducted in the eastern coast of Ghana also found that well-developed, local fuelwood markets motivates mangrove harvesters to prioritize short-term economic gains over long-term ecological benefits from mangrove protection [70]. Similarly, evidence from the Cameroon estuary indicates that growing wood markets in nearby cities is a major cause of mangrove forest overexploitation, as the sale of mangrove wood is a major source of household incomes [69]. In the study region, urbanization was evidenced by population changes between Essiama and Atuabo and related land use/land cover conversions to artificial/bare areas. Such changes were also influenced by elevation. The aforementioned locations are the centers for oil and gas infrastructure development. Urbanization trends in this area also account for mangrove forest losses.

Mangrove ecosystems store large quantities of carbon in below- and aboveground components, hence, if disturbed, will result in the release of high levels of greenhouse gas emissions [57]. In Ghana, mangroves are of interest for inclusion in climate mitigation strategies. The mean aboveground carbon storage potential of the coastal landscapes in Southwestern Ghana was estimated at 114.66 Mg C ha$^{-1}$ (Supplementary Table S2). This lies within the reported range from 5.2 to 312 Mg C ha$^{-1}$ aboveground carbon pools for mangroves in West-Central Africa [71]. Considering the total extent of the landscape, the aboveground mangrove carbon storage potential ranged from 6.6 to 87,196.2 Mg C. This potential showed a decreasing trend over the study period (Supplementary Table S2). Additionally, there were spatial variations in the distribution patterns of potential carbon storage in the study landscape. Generally, areas along the estuaries of the Ankobra and Amanzule rivers recorded relatively higher potential carbon storage compared to areas farther away. Global studies reporting on sites in West Africa found direct correlations between spatial variations in mangrove carbon storage potential and the geomorphic positions of mangrove stands [59]. Other environmental factors such as soil properties, salinity and precipitation also influence mangrove carbon storage [71].

Contrary to mangroves, aboveground rubber carbon storage potential showed increasing temporal trends and ranged from 0.05 to 429 Mg C. Despite the increasing carbon storage potential and the relatively large extent of rubber plantations, mangrove carbon storage potential was two hundred-fold greater than rubber carbon storage during the study period. This finding reinforces the evidence for investments in mangrove blue carbon as a viable option for achieving climate change mitigation targets in tropical coastal landscapes [72].

### 4.2. Implications for Land Use Planning

Within the context of Ghana's three-tiered land use planning approach, spatial development frameworks are prepared at the strategic level, which in turn guide the preparation of structure and local plans at lower tiers [73,74]. This approach to land use planning hinges on setting goals and objectives and developing future scenarios as a basis for expression and implementation of spatial social, environmental and economic policies. The spatial development framework for the coastal subregion of Ghana's western region identifies the need to reconcile industrial development and conservation of sensitive coastal habitats. Yet this spatial plan falls short of identifying areas within the landscape for the supply of critical ES as a basis for conservation decisions. Tradeoffs between land use for agri-

culture, industrial development and conservation of coastal ecosystems will likely arise with the current pace of land transformation dominated by market-oriented drivers such as increasing monoculture of rubber plantations, mining activities and oil and gas infrastructure development [63,64]. Moreover, tradeoffs in the supply of relevant ES will also emerge and therefore require prioritization of ES supply. In the study region, tradeoffs exist between mangrove conservation as a regulating service and mangrove fuelwood supply as provisioning services. Similarly, understanding the carbon storage potential of rubber plantation versus mangrove forests will support prioritization of land use investments for implementation of climate mitigation strategies. Understanding and addressing these tradeoffs in a rapidly transforming land system requires integration of ES perspectives into the planning process [21]. Comparative analysis of the quantities of ES supply will support pragmatic objectives and decisions regarding areas where critical ES supply require maintenance and also areas where tradeoffs can be minimized. Prime areas for agriculture can be maintained considering biophysical attributes supporting food production. This will also strengthen biophysical justifications for land use scenarios presented during the planning process while proactively supporting decisions to protect ecosystems.

Additionally, development of zoning regulations as part of structure plan preparation processes will benefit from landscape scale estimations of ES. Areas of high biodiversity and ES will be apparent, leading to their protection through the use of appropriate regulatory instruments. Areas where there are risks due to tensions between conservation and coastal development can be identified and mitigated.

*4.3. Study Limitations*

Despite the similarities between the ecosystems from which data were transferred and that of the study area, the application of benefit transfer in this study risked introduction of errors due to landscape heterogeneity. This implies that transfer sites were not truly representative of the corresponding sites in the study landscape. This is because within each broad land cover class there are subclasses with different attributes. Thus, implementation of benefit transfer in this study assumed that all land cover units within a broad land cover class are the same. Moreover, key variations in ES supply were not evident due to the coarse temporal resolution of ten-year intervals of the remote sensing datasets. Finally, results of the study were not validated to confirm or not confirm, the estimates of ES values in the study region. Despite these challenges, the applied methodology provides a quick first step toward quantifying, mapping and including information on ES into coastal land use planning in the study region.

*4.4. Conclusions and Future Outlook*

Mapping and assessing ES require data at the appropriate scale and spatial and temporal resolutions [19]. In data-poor regions, such as Southwestern Ghana, the absence of such datasets hinders ES mapping and valuation. However, availability of data collected using primary studies and agricultural production statistics compiled by government institutions presents opportunities for mapping, especially, relevant regulating and provisioning ES. This study demonstrated the use of remote sensing data, cassava and plantain yield statistics and estimates of mangrove forest stand biomass and aboveground carbon in rubber plantation to quantify the landscapes' potential to supply ES. Relevant ES supply potentials quantified were food supply, fuelwood supply and carbon storage, and their spatial distribution patterns. The mean aboveground carbon storage potential of the coastal landscapes in Southwestern Ghana was estimated at 114.66 Mg C ha$^{-1}$. However, considering the total extent of the landscape, the aboveground mangrove carbon storage potential ranged from 6.6 to 87,196.2 Mg C with spatial and temporal variations in the distribution of mangrove carbon storage potential. Relatedly, mangrove fuelwood supply potential ranged from 0.01 to 87.19 tons and varied over the study period and across different locations on the landscape. Overall, potential to supply mangrove fuelwood and also aboveground carbon storage in mangrove ecosystems depicted decreasing trends. The potential for

food supply from cropland and carbon storage in rubber plantation increased during the study period. Cassava supply ranged from 1.7 to 23.4 gigatons and plantain supply ranged from 0.001 to 3.3 gigatons. Rubber carbon storage potential ranged from 0.05 to 429 Mg C. Population, slope and elevation exhibited strong effects on LULC conversions to food crop and rubber plantations, whereas these factors were less important in determining mangrove forest conversions.

Rapid transformation of the land system in the study region is a major risk to sustainable supply of ES and to the minimization of tradeoffs in land use decision-making. Integration of ES perspectives will strengthen the biophysical basis of land use planning and decision-making in the region. Future ES mapping should take into account estimation of regional balance in food supply, as this will be necessary for optimal allocation of land for food production and conservation of critical coastal ecosystems.

**Supplementary Materials:** The following supporting information can be downloaded at: https://www.mdpi.com/article/10.3390/land11091408/s1, Table S1: Regional food-crop production statistics; Table S2: Mangrove carbon estimates from selected forest stands; Table S3: Mangrove biomass estimates from selected forest stands; Table S4: Rubber carbon estimates from selected plots.

**Author Contributions:** Conceptualization, S.K.; methodology, S.K.; software, A.O. and S.K.; formal analysis, S.K., A.O. and J.N.I.; writing—original draft preparation, S.K.; writing—review and editing, C.F. and J.N.I.; supervision, C.F.; funding acquisition, C.F. All authors have read and agreed to the published version of the manuscript.

**Funding:** This research was funded by the German Federal Ministry of Environment and Research (BMBF) through a long term EU-Africa research and innovation Partnership on food and nutrition security and sustainable Agriculture (LEAP-Agri) under the grant number at Martin Luther University Halle-Wittenberg [01DG18020].

**Institutional Review Board Statement:** Not applicable.

**Informed Consent Statement:** Not applicable.

**Data Availability Statement:** Not applicable.

**Acknowledgments:** We acknowledge the support of Hen Mpoano (Our Coast) in providing access to spatial datasets for this study. We express our appreciation to the Regional Director of the Ministry of Food and Agriculture (MoFA), Western Region, for sharing regional agricultural statistics. Messrs. Daniel Nii Doku Nortey, Justice Mensah and Joshua Adotey are duly acknowledged for their support in providing data for this study. We also thank the four anonymous reviewers for their helpful comments and suggestions on a draft version of this manuscript.

**Conflicts of Interest:** The authors declare no conflict of interest.

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
