# Peer review of "Implications of Spatio-Temporal Land Use/Cover Changes for Ecosystem Services Supply in the Coastal Landscapes of Southwestern Ghana, West Africa"

_land, doi:10.3390/land11091408_

Round 1

Reviewer 1 Report

The study ‘’ Spatio-Temporal LULCC and Its Implications for Ecosystem Services Supply in the Coastal Landscapes of Southwestern Ghana, West Africa’’ is very good. I think the paper is interesting and it discusses the land cover changes and implication Ecosystem Services Supply. However, I identified the following serious concerns that should be taken into account before acceptance of the manuscript.

The text of this paper in general needs a thorough review, as there are multiple spelling and grammatical errors.

Abstract in general: The abstract should focus on the summary of the study, main findings, and possible implications.

Line 242; Table s1 ? Line 618 table s2? Please write the only table 2 and table 2 in main text.

Line 79-82; give the suitable reference, More research background and motivation should be added to the Introduction section. Although, I propose some new papers must be added in the reference list and text which will also help you to make it more intriguing such. https://doi.org/10.1038/s41598-022-17454-y, https://doi.org/10.1016/j.envdev.2020.100576,  doi:10.3390/ijerph14080840,

Line 177; Give the link of USGS where you download Landsat images

Line 197-199; Need a piece of detailed information about the atmospheric correction

Discussion: As per the instruction given by the journal “The findings and their implications should be discussed in the broadest context possible and the limitations of the work highlighted”

The conclusion is too general. What are the key findings of this study?

Overall, the study conducted is interesting but a minor revision of the entire manuscript is essentially required for publication in this journal. Hence, I recommend reconsideration after a minor revision of the manuscript.  

Author Response

Point 1: The text of this paper in general needs a thorough review, as there are multiple spelling and grammatical errors

Response 1: Thanks. We have given the manuscript to a native speaker for proofreading. Consequently, all grammatical and spelling errors have been corrected throughout the manuscript. 

Point 2: Abstract in general: The abstract should focus on the summary of the study, main findings, and possible implications.

Response 2: Thanks. The abstract has been revised as follows;

Land use/land cover change (LULCC) is an important driver of ecosystem changes in coastal areas. Despite being pervasive in coastal Ghana, LULCC has not been investigated to understand its effects on the potential for coastal landscapes to supply ecosystem services (ES). In this study, the impacts of LULCC on the potential supply of ES by coastal landscapes in Southwestern Ghana was assessed for the years, 2008 and 2018, using remote sensing and benefit transfer approaches. Based on available data, relevant provisioning and regulating ES were selected for the assessment while indicators to aid the quantification of the ES were obtained from literature. Supervised classification methods and maximum likelihood algorithms were used to prepare land use/land cover (LULC) maps and the derived LULC categories were assigned according to the descriptions of the Land Cover Classification System (LCCS). Potential supply of provisioning (food, fuel wood) and regulating (carbon storage) services was quantified and the spatial and temporal distributions of these ES illustrated using maps. The results show variations in food and fuel wood supply and carbon storage potentials over the study period and across different locations on the landscape. Potentials for fuel wood supply and carbon storage in mangrove forests indicated declining trends between 2008 and 2018. On the other hand, food crop supply and carbon storage potential in rubber plantations depicted increasing patterns over the same period. Population, slope and elevation exhibited strong effects on LULC conversions to food crop and rubber plantations whereas these factors were less important in determining mangrove forest conversions. The findings of the study have implications for identifying and addressing tradeoffs between land uses for agriculture, industrial development and conservation of critical coastal ES within the context of rapid land system transformations in the study region.   

Point  3: Line 242; Table s1 ? Line 618 table s2? Please write the only table 2 and table 2 in main text.

Response 3: Thanks. These are references to supplementary materials. S1, S2 etc. have been fully spelt out. 

Point 4: Line 79-82; give the suitable reference, More research background and motivation should be added to the Introduction section. Although, I propose some new papers must be added in the reference list and text which will also help you to make it more intriguing such. https://doi.org/10.1038/s41598-022-17454-y, https://doi.org/10.1016/j.envdev.2020.100576, doi:10.3390/ijerph14080840,

Response 4: Thanks for the helpful suggestions. The suggested literature and others have been consulted and woven into the arguments in the introdcution and discussion sections. They are now found in the bibliography list as numbers, 9 and 10. Furthermore, more background and context for the research focusing on the direct and indirect drivers of LULCC have been included. In the revised manuscript, these can be found under lines 48- 57.

Point 5: Line 177; Give the link of USGS where you download Landsat images

Response 5: USGS source  (https://earthexplorer.usgs.gov/) has been included. This can be found in Table 1. 

Point 6: Line 197-199; Need a piece of detailed information about the atmospheric correction

Response 6: Thanks for your suggestion. However, the corrections we effected were mostly radiometric and geometric. We dd not have to do any major atmospheric correction because the images were cloud-free and not too noisy

Point 7: Discussion: As per the instruction given by the journal “The findings and their implications should be discussed in the broadest context possible and the limitations of the work highlighted”. 

Response 7:Thanks. Additional paragraphs have been inserted to broaden the discussion. In the revised manuscript these can be found under lines 596 – 602. Limitations of the study have also been highlighted under lines 661 – 684. 

Point 8: The conclusion is too general. What are the key findings of this study?

Response 8: Thanks. specific key findings of the study have been incorporated under 714 -725 in the revised manuscript.  

Reviewer 2 Report

The authors have selected an important topic of ecosystem services particularly in the context of coastal areas. It always remained under debate how to define and quantify the actual extent of ecosystem services. In the context, I appreciate the effort of the authors to address the aspects of basic services in terms of food and fuel wood, and service related to environmental pollution / climate mitigation through carbon sequestration. These are important variables particularly in the case of Ghana. However, the actual scope of ecosystem services in tangible form has yet to be explored further for which advance research can be done based on the findings of this paper. Although there can a long list of items for the improvement of this paper, I have following key observations based on technical aspects of the subject, particularly related to the valuation of the ecosystem services:

1.     The relationship between the domestic utilization of fuel wood and economic aspect of rubber  plants vis-à-vis LULCC due to land replacement factor vis-à-vis Rubber crop oriented extensification and carbon sequestration cannot be overlooked. It needs to be emphasized not only at appropriate place in introduction but also in discussion for the results and implications of the study.

2.     Results’ Sub-section 3.2.3 Carbon Storage – it will be good if CO2eq value be figured ot and incorporated against the reported Carbon storage values for Mangrove and Rubber plantation. Through valuation, it will help to see potential earning through carbon offsets / credit schemes, which is the actual tangible economic form of ecosystem services. Further, a LULCC comparison should also be added in discussion and implication sections regarding the crop rotation aspect vis-a-vis the fluctuations in carbon storage by Rubber including its positive or negative correlation with Mangroves over time and again.

Author Response

Point 1: The relationship between the domestic utilization of fuel wood and economic aspect of rubber  plants vis-à-vis LULCC due to land replacement factor vis-à-vis Rubber crop oriented extensification and carbon sequestration cannot be overlooked. It needs to be emphasized not only at appropriate place in introduction but also in discussion for the results and implications of the study.

Response 1: Thanks. These relationships have been further explained in the introduction. Below is an excerpt from the introduction in the revised manuscript which  addresses this comment. 

“Increasingly, land losses from oil palm, cropland and shrubland favor gains in rubber plantation [34]. Nonetheless rubber plantations are fragmented over the landscape as their establishment on few acres of land are determined by individual land owners in the context of an outgrower scheme [34]”

Point 2. Results’ Sub-section 3.2.3 Carbon Storage – it will be good if CO2eq value be figured out and incorporated against the reported Carbon storage values for Mangrove and Rubber plantation. Through valuation, it will help to see potential earning through carbon offsets / credit schemes, which is the actual tangible economic form of ecosystem services.  Further, a LULCC comparison should also be added in discussion and implication sections regarding the crop rotation aspect vis-a-vis the fluctuations in carbon storage by Rubber including its positive or negative correlation with Mangroves over time and again.

Response 2: Thanks for this suggestion. However, the study methodology focussed on quantification of ecosystem service supply potentials of the landscape in biophysical units.  

Comparison of LULCC considering rubber and mangrove have been included in the discussion. However, the comment regarding crop rotation aspects is unclear. Our LULC transitions show that mangrove forests are not converted to rubber plantations and vice versa. This is because mangroves thrive in brackish water and low elevation areas whereas rubber thrives in relatively higher elevations. 

Reviewer 3 Report

Dear authors,

the topic of this manuscript is very interesting and relevant. In fact, I believe this manuscript has scientific soundness and could be a helpful work to move forward in this thematic scientif field.

However, before the publication the authors should revise the use of the English language as well as to solve some minor issues - i.e., avoid to use acronyms on the title of the manuscript.

Author Response

Point 1: Before the publication the authors should revise the use of the English language as well as to solve some minor issues - i.e., avoid to use acronyms on the title of the manuscript. 

Response 1: Thanks. We have given the manuscript to a native speaker for proofreading. Consequently, all grammatical and spelling errors have been corrected throughout the revised manuscript. Acronym is also removed from the title.

Reviewer 4 Report

In the scientific literature, land use/land cover is often used instead of landuse/landcover. I kindly ask the authors to adhere to the terminology used in the scientific literature.

In the Introduction, I think it would have been useful to further develop the relationship between land use/land cover changes worldwide and their implications for Ecosystem Services Supply. This matter is topical and there is an extensive literature that deals with the relationship between land use/land cover changes and different global processes that include ecosystem services that deserve to be mentioned and addressed.

The description of the study area seems insufficient in the context of the present subject. Some more references on land use/land cover in Ghana and the impact of land use/land cover changes the on ecosystems and the services they provide would also be useful.

The Methodological Framework is clearly presented. The Results are comprehensive and useful.

Also, the paper presents a series of Discussions that are very useful in the context of the results obtained, both from a scientific and applied point of view. In this sense, it also brings a series of specifications related to the implications of the study for land use planning, which gives it an even more applied message.

Author Response

Point 1: In the scientific literature, land use/land cover is often used instead of landuse/landcover. I kindly ask the authors to adhere to the terminology used in the scientific literature. 

Response 1: Thanks. Landuse/landcover has been replaced with land use/land cover throughout the revised manuscript

Point 2: In the Introduction, I think it would have been useful to further develop the relationship between land use/land cover changes worldwide and their implications for Ecosystem Services Supply. This matter is topical and there is an extensive literature that deals with the relationship between land use/land cover changes and different global processes that include ecosystem services that deserve to be mentioned and addressed.

Response 2: Thanks. The relationship between land use/cover dynamics and ES at the global scale has been incorporated in the introduction section.  These can be found under lines 48 – 57 in the revised manuscript as follows;

"Global scale assessments estimate annual ES losses due to land use changes at approximately $20.2 trillion [11]. Increasingly, individuals and societies respond to opportunities created by globalization processes including market conditions by altering land uses [12,13]. These changes trigger degradation and conversions of high-value ecosystems such as forests, cropland, water and grasslands to low-value land uses [10,11]. Over the past couple of decades, ecosystem degradation have heigthened due to an exponential increase in population and doubling of economic activities with attendant increase in the demand for ecosystem goods and serivces [2]. In many developing countries, weak or absent land use regulatory institutions are critical among the conditions giving rise to rapid modifications of ecosystems and landscapes [13]".

Point 3:The description of the study area seems insufficient in the context of the present subject. Some more references on land use/land cover in Ghana and the impact of land use/land cover changes the on ecosystems and the services they provide would also be useful.

Response 3: Thanks for the suggestion. Additional information have been provided to deepen the study area description. These can be found in the revised manuscript under lines 147 – 158 as follows; 

"The discovery of oil and gas in commercial quantities off the continental shelf in Southwestern Ghana, ushered the region into a new wave of competition between industrial, residential and agricultural land uses [36]. This is manifested by the losses of farmland and forests in favor of built-up areas in the region’s urban core and peripheries [36,45]. It is noteworthy that onshore oil and gas infrastructure is expanding into ecologically sensitive areas of this landscape, thereby causing further habitat fragmentation and threatening wildlife [46]. The population has doubled over the last decade and is increasing above the national average due to the region being a focal point for in-migration [45]. Historically, economic development in this area was driven by a vibrant fishing industry, but more recently, the fisheries sector has suffered decline [47]. Like other coastal regions in Ghana, the well-being of the local population are inexorably linked to natural resources which underpin their contentment with ES [48]".